# FEW-SHOT INCREMENTAL LEARNING USING HYPERTRANSFORMERS

## ABSTRACT

Incremental few-shot learning methods make it possible to learn without forgetting from multiple few-shot tasks arriving sequentially. In this work we approach this problem using the recently published HyperTransformer (HT): a hypernetwork that generates task-specific CNN weights directly from the support set. We propose to re-use these generated weights as an input to the HT for the next task of the continual-learning sequence. Thus, the HT uses the weights themselves as the representation of the previously learned tasks. This approach is different from most continual learning algorithms that typically rely on using replay buffers, weight regularization or task-dependent architectural changes. Instead, we show that the HT works akin to a recurrent model, relying on the weights from the previous task and a support set from a new task. We demonstrate that a single HT equipped with a prototypical loss is capable of learning and retaining knowledge about past tasks for two continual learning scenarios: incremental-task learning and incremental-class learning.

## 1 INTRODUCTION

Incremental few-shot learning combines the challenges of both few-shot learning and continual learning together: it seeks a way to learn from very limited demonstrations presented continually to the learner. This combination is desirable since it represents a more genuine model of how biological systems including humans acquire new knowledge: we often do not need a large amount of information to learn a novel concept and after learning about it we retain that knowledge for a long time. In addition, achieving this would dramatically simplify learning of important practical applications, such as robots continually adapting to a novel environment layout from an incoming stream of demonstrations. Another example is privacy-preserving learning, where users run the model sequentially on their private data, sharing only the weights that are continually absorbing the data without ever exposing it.

We focus on a recently published few-shot learning method called HYPERTRANSFORMER (HT; Zhmoginov et al. 2022), which trains a large hypernetwork (Ha et al., 2016) by extracting knowledge from a set of training few-shot learning tasks. The HT is then able to directly generate weights of a much smaller Convolutional Neural Network (CNN) model focused on solving a particular task from just a few examples provided in the support set. It works by decoupling the task-domain knowledge (represented by a transformer; Vaswani et al. 2017) from the learner itself (a CNN), which only needs to know about the specific task that is being solved.

In this paper, we propose an INCREMENTAL HYPERTRANSFORMER (IHT) aimed at exploring the capability of the HT to *update the CNN weights* with information about new tasks, while retaining the knowledge about previously learned tasks. In other words, given the weights $\theta_{t-1}$ generated after seeing some previous tasks $\{\tau\}_{\tau=0}^{t-1}$ and a new task $t$, the IHT generates the weights $\theta_t$ that are suited for all the tasks $\{\tau\}_{\tau=0}^{t}$.

In order for the IHT to be able to absorb a continual stream of tasks, we modified the loss function from a cross-entropy that was used in the HT to a more flexible prototypical loss (Snell et al., 2017). As the tasks come along, we maintain and update a set of prototypes in the embedding space, one for each class of any given task. The prototypes are then used to predict the class and task attributes for a given input sample.

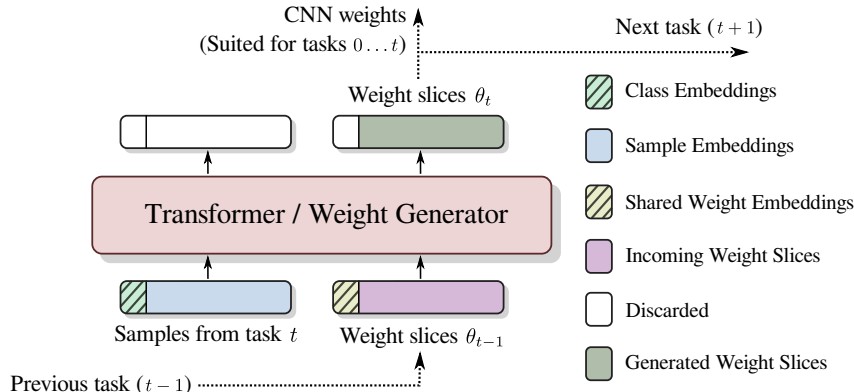

Figure 1: The information flow of the IHT. In the original HT each of the input weight embeddings are initialized with an empty weight slice. Our proposal is to pass weight slice information from previously learned tasks as an input to the new iteration of the HT.

The IHT works in two different continual learning scenarios: task-incremental learning (predict class attribute using the task information) and class-incremental learning (predict both class and task attributes). Moreover, we show empirically that a model trained with the class-incremental learning objective is also suited for the task-incremental learning with performance comparable to the models specifically trained with a task-incremental objective.

We demonstrate that models learned by the IHT do not suffer from catastrophic forgetting. Moreover, in some smaller models we even see cases of positive backward transfer, where the performance on a given task actually improves for subsequently generated weights.

Since the IHT is trained to optimize all the generated weights $\{\theta_\tau\}_{\tau=0}^T$ together, the model can be preempted at any point $\tau \leq T$ during the inference with weights $\theta_\tau$ suited for any task $0 \leq p \leq \tau$. Moreover, we show that the performance of the model improves for all the generated weights when the IHT is trained on more tasks $T$. We also designed the IHT to work as a recurrent system and its parameters are independent from a given step. Therefore, it can be continued and applied beyond the number of tasks $T$ it was trained for.

## 2 RELATED WORK

**Few-shot learning**    Many of few-shot learning methods fall into one of two categories: metric-based learning and optimization-based learning. First, metric-based learning methods (Vinyals et al., 2016; Snell et al., 2017; Sung et al., 2018; Oreshkin et al., 2018) train a fixed embedding network that works universally for any task. The prediction is then based on the distances between the labeled and query samples in that embedding space. These methods are not specifically tailored for the continual learning problem, since they treat every task independently and have no memory of the past tasks. In contrast to this method, our proposed IHT can be seen as an "adaptive" metric-based learner, where the weights $\theta_t$ are changing to adapt better to the task $t$ and retain the knowledge of the past tasks.

Second, optimization-based methods (Finn et al., 2017; Nichol & Schulman, 2018; Antoniou et al., 2019; Rusu et al., 2019) consisting of variations of a seminal MAML paper propose to learn an initial fixed embedding, that is later adapted to a given task using few gradient-based steps. All by themselves these methods are not able to learn continually, since naively adapting for a new task will result in a catastrophic forgetting of the previously learned information.

**Continual learning.**    We believe that compared to the related work (see Biesialska et al. 2020 for an overview), our approach requires the least conceptual overhead, since it does not add additional constraints to the method beyond the weights generated from the previous task. In particular, we do not inject replay data from past tasks (Lopez-Paz & Ranzato, 2017; Riemer et al., 2018; Rolnick et al., 2019; Wang et al., 2021a), do not explicitly regularize the weights (Kirkpatrick et al., 2017; Zenke et al., 2017) and do not introduce complex graph structures (Tao et al., 2020; Zhang et al., 2021), data routing or any other architectural changes to the inference model (Rusu et al., 2016). Instead, we reuse the same principle that made HYPERTRANSFORMER work in the first

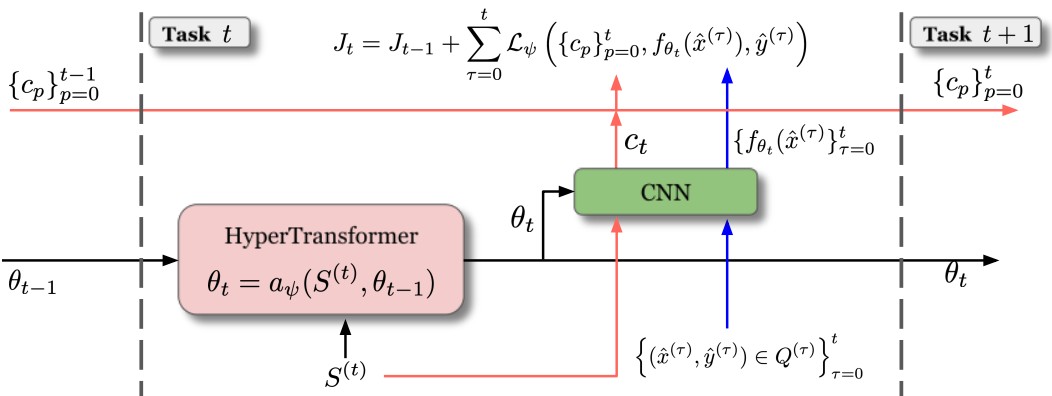

Figure 2: The diagram of few-shot continual learning with INCREMENTAL HYPERTRANSFORMER. For each new task $t$, the IHT uses the support set $S^{(t)}$ and previously generated weights $\theta_{t-1}$ to generate new weights $\theta_t$. The support set is then passed through those weights to compute the prototypes $c_t$ and update the set of prototypes for all the tasks $\{c_p\}_{p=0}^t$. Finally, we pass the query set of every task $\{Q^{(\tau)}\}_{\tau=0}^t$ to we evaluate the quality of the generated weights.

place: decouple a small CNN classification model re-generated for each incoming task from a large Transformer model that generates CNN weights, in effect learning how to adapt them to a new incoming task in a way that avoids forgetting prior knowledge. In this regard, the closest analogy to our approach would be slow and fast weights (Munkhdalai & Yu, 2017), with the IHT weights being analogous to the slow weights that accumulate the knowledge and generate CNN weights as fast weights.

**Incremental few-shot learning.** Prior work on incremental few-shot learning frequently uses a similar fast/slow weight separation, but the fast weights are typically generated using some form of gradient-based optimization (Ren et al., 2019; Lee et al., 2021). Furthermore, many incremental few-shot learning techniques rely on frozen feature extractors (Ren et al., 2019; Perez-Rua et al., 2020; Wang et al., 2021b; Zhang et al., 2021), which could potentially hinder their performance in situations where the input domain changes considerably between tasks. Some of these techniques (Wang et al., 2021b) are attention-based and can be viewed as special cases of our approach, where most of the model layers are trained or pretrained instead of being generated.

Many methods exist that try to specifically address the catastrophic forgetting by regularizing either the weights directly by restricting the update of some of them (Mazumder et al., 2021; Chen & Lee, 2020) or by constrain the optimization procedure itself (Shi et al., 2021; Gupta et al., 2020).

## 3 HYPERTRANSFORMER ARCHITECTURE

We consider a few-shot learning scenario, where we are given a set of tasks $\{t|t \in \mathcal{T}\}$ split into $\mathcal{T}_{\text{train}}$ and $\mathcal{T}_{\text{test}}$. Each task $t$ is specified via a $K$-way $N$-shot support set $S^{(t)} = (x_i^{(t)}, y_i^{(t)})_{i=1}^{NK}$ and a query set $Q^{(t)} = (\hat{x}_i^{(t)}, \hat{y}_i^{(t)})_{i=1}^{\hat{N}K}$, where $K$ is the number of classes in the episode, $N$ is the number of labeled demonstrations of each class, and $\hat{N}$ (typically $\hat{N} \gg N$) is the number of query samples to be classified.

At each training iteration, a new episode is sampled without replacement from $\mathcal{T}_{\text{train}}$. The support set from that episode is used by the HT (parameterized by $\psi$) to directly generate the weights of the CNN via $\theta_t = a_\psi(S^{(t)})$. The loss is computed by passing the query set $(\hat{x}, \hat{y})$ from the same task through the generated CNN $f_{\theta_t}(\hat{x})$ and using a cross-entropy objective. After the weights $\psi$ of the HT are trained, the tasks are sampled from a separate set of $\mathcal{T}_{\text{test}}$ to evaluate its performance.

At the core of the HT lies an attention mechanism that generates CNN weights for a given layer from the support set. The transformer receives the embedding of the support set and placeholders for weight embeddings *initialized with zero weight slices* (see Fig. 1). At the output the support set embedding is ignored and the generated weight slices are converted into weights of the CNN.

# 4 INCREMENTAL HYPERTRANSFORMER

The HT can naturally extended the HT architecture to a continual stream of tasks by passing the generated weights from already learned tasks as input weight embeddings into the weight generator for a new task (see Fig. 1). This way the learned weights themselves act as both the input and the output of the the IHT, performing a dual function: storing information about the previous tasks *as well as* serving as CNN weights for evaluation on already seen tasks.

Fig. 2 demonstrates the overall learning flow of the proposed algorithm. During the training, $T$ episodes are sampled sequentially at random without replacement. For each task $t$, the IHT receives the support set of that task $S^{(t)}$ as well as the previous weights $\theta_{t-1}$[1] and generates the weights

$$\theta_t = a_\psi(S^{(t)}, \theta_{t-1}), \tag{1}$$

that are suited for all the tasks $\{\tau\}_{\tau=1}^t$. Therefore, for each step $t$ we want to minimize the loss on the query sets of every task up to $t$:

$$J_t(\psi) = \sum_{\tau=0}^t \mathcal{L}_\psi \left( f_{\theta_t}(\hat{x}^{(\tau)}), \hat{y}^{(\tau)} \right). \tag{2}$$

The final loss function is given by the sum of the losses for all the tasks:

$$\arg\min_\psi \sum_{t=0}^T J_t(\psi). \tag{3}$$

The IHT sequentially generates a series of weights $\{\theta_\tau\}_{\tau=0}^t$, each of which are suited for all the tasks coming before it: $\theta_0$ performs well only on the task $\mathcal{T}_0$, $\theta_1$ performs well on the tasks $\mathcal{T}_0$ and $\mathcal{T}_1$, and so forth. This design choice allows us to have a "preemptive" continual learning setup, where the IHT trained for $T$ tasks can be stopped at any point at $\tau < T$ and still have well-performing weights $\theta_\tau$ for all the tasks seen so far. An alternative would be a setup where a user knows the exact number of tasks in the sequence and only cares about the performance after the final task $T$. This would correspond to minimizing only the last term $J_T$ in (3). We have experimented with this version and did not observe a noticeable improvement over the one we propose above.

Another desirable property of the proposed IHT architecture is its recurrence. The HT parameters do not depend on the task information and simply take the $\theta$ and the support set as its input. Therefore, it is possible not only to preempt IHT by stopping early, but also extend the trained model to continue generating weights for more tasks that it has been trained on. We will demonstrate this ability in the experimental section.

## 4.1 PROTOTYPICAL LOSS

The last element of the algorithm that we have left to discuss is the exact form of loss function $\mathcal{L}_\psi()\cdot$ in (2). The original HT used the cross-entropy loss, which is not well suited for continual learning, since the number of labels that it predicts is tied to the number of parameters in the head layer of the $\theta$. As the number of tasks increases, the architecture of CNN needs to be adjusted, which goes against our design principle of a recurrent IHT architecture. Another alternative would be to fix the head layer to the $K$-way classification problem across all the tasks and predict only the class information, ignoring the task attributes (which is known as domain-incremental learning Hsu et al., 2018). However, in this case the classes with the same label attribute from different tasks would be minimized to the same spot in the embedding space, creating a collision. Moreover, since these label attributes do not carry any semantic meaning and are drawn at random for every episode, the collisions would happen at random and the IHT would not be able to adjust or learn from them. In the Appendix A.1, we show that the accuracy of this setup drops dramatically as the number of tasks increases and this method becomes impractical right away for as little as two tasks.

To make the method usable we need to decouple the class predictions of every task while keeping the overall dimensionality of the embedding space fixed. One way to solve this problem is to come up with a fixed arrangement of $TK$ points, but any kind of such arrangement is sub-optimal since it is impossible to place $TK$ points equidistant from each other in a fixed dimensional space for large enough $T$. A much more elegant solution would be to learn the location of these class prototypes

---

[1]The initial weights $\theta_0$ are generated using empty weight slices as it is done in the original HT.

from the support set itself, e.g. with a prototypical loss (Snell et al., 2017). The prototypes are computed by averaging the embeddings of support samples from a given class $k$ *and* task $\tau$:

$$c_{\tau k} = \frac{1}{N} \sum_{(x,y) \in S^{(\tau)}} f_{\theta_\tau}(x) \mathbf{1}_{y=k}. \tag{4}$$

We can use the prototypes in two different continual learning scenarios. First, for the *task-incremental learning*, we are assumed to have access to the task we are solving and need to predict only the class information. The probability of the sample belonging to a class $k$ given the task $\tau$ is then equal to the softmax of the *l2* distance between the sample and the prototype normalized over the distances to the prototypes from all the classes from $\tau$:

$$p(\hat{y} = k | \hat{x}, \tau) = \frac{\exp(-\|f_{\theta_t}(\hat{x}) - c_{\tau k}\|^2)}{\sum_{k'} \exp(-\|f_{\theta_t}(\hat{x}) - c_{\tau k'}\|^2)}. \tag{5}$$

Second, for the more general *class-incremental learning*, we need to predict both task and class attributes. The probability of the sample belonging to class $k$ of task $\tau$ is equal to the softmax of the *L2* distance between the sample and the prototype normalized over the distances to the prototypes from all the classes and all the tasks:

$$p(\hat{y} = k, \tau | \hat{x}) = \frac{\exp(-\|f_{\theta_t}(\hat{x}) - c_{\tau k}\|^2)}{\sum_{\tau' k'} \exp(-\|f_{\theta_t}(\hat{x}) - c_{\tau' k'}\|^2)}. \tag{6}$$

The final loss function is given by minimizing the negative log probability of the chosen softmax over the query set. Algorithm 1 describes the procedure to compute that loss value.

Empirically, we noticed that the IHT models trained with the class-incremental learning objective (6) are applied equally well to both class-incremental and task-incremental settings, while models trained with the task-incremental objective (5) work well only in the task-incremental setting and rarely better than the models trained using (6). In what follows we are going to focus only on the IHT trained with (6), but evaluate them for both task- and class-incremental learning scenarios.

Notice that the prototypes are computed with respect to the current weights $\theta_\tau$ in (4) for the task $\tau$, however they need to be used down the line to compare the embeddings produced by subsequent weights $\theta_t$ in (6). Ideally, once the new weights $\theta_t$ are generated, the prototypes need to be recomputed as well. However, in the true spirit of continual learning, we are not supposed to re-use the support samples after the task has been already processed. We found out that freezing the prototypes after they are computed provides a viable solution to this problem. We noticed that compared to recomputing the prototypes every step, this gives only a marginal difference in performance.

We do want to note an important use-case where recomputing the prototypes might still be possible and even desirable. Notice that the weights $\theta_t$ are not affected by this issue and are computed in a continual learning manner from (1) without the use of the information from the previous task. The support set is needed only to update the prototypes through generated weights, which is a relatively cheap operation. Thus, we can envision a privacy-preserving scenario in which the weights are updated and passed from client to client in a continual learning way and the prototypes needed to "unlock" those weights belong to the clients that hold the actual data.

## 5 Connection Between Prototypical Loss and MAML

While the core idea behind the prototypical loss is very natural, this approach can also be viewed as a special case of a simple 1-step MAML-like learning algorithm. This can be demonstrated by considering a simple classification model $\boldsymbol{q}(x; \phi) = s(\boldsymbol{W} f_\theta(x) + \boldsymbol{b})$ with $\phi = (\boldsymbol{W}, \boldsymbol{b}, \theta)$, where $f_\theta(x)$ is the embedding and $s(\cdot)$ is a softmax function. MAML algorithm identifies such initial weights $\phi^0$ that any task $\tau$ with just a few gradient descent steps initialized at $\phi^0$ brings the model towards a task-specific local optimum of $\mathcal{L}_\tau$.

Notice that if any label assignment in the training tasks is equally likely, it is natural for $\boldsymbol{q}(x; \phi^0)$ to not prefer any particular label over the others. Guided by this, let us choose $\boldsymbol{W}^0$ and $\boldsymbol{b}^0$ that are *label-independent*. Substituting $\phi = \phi^0 + \delta\phi$ into $\boldsymbol{q}(x; \phi)$, we then obtain

$$q_\ell(x; \phi) = q_\ell(x; \phi^0) + s'_\ell(\cdot) \left( \delta \boldsymbol{W}_\ell f_{\theta^0}(x) + \delta b_\ell + \boldsymbol{W}_\ell^0 \frac{\partial f}{\partial \theta}(x; \theta^0) \delta\theta \right) + O(\delta\phi^2),$$

---

**Algorithm 1** Incremental-class learning using HYPERTRANSFORMER with Prototypical Loss.

---

**Input:** $T$ randomly sampled $K$-way $N$-shot episodes: $\{S^{(t)}; Q^{(t)}\}_{t=0}^T$.
**Output:** The loss value $J$ for the generated set of tasks.

  1:  $J \leftarrow 0$                                                           ▷ Initialize the loss.
  2:  $\theta_{-1} \leftarrow 0$                                                         ▷ Initialize the weights.
  3:  **for** $t \leftarrow 0$ to $T$ **do**
  4:     $\theta_t \leftarrow a_\psi(S^{(t)}, \theta_{t-1})$                      ▷ Generate weight for current task.
  5:     **for** $k \leftarrow 0$ to $K$ **do**       ▷ Compute prototypes for every class of the current task.
  6:         $c_{tk} \leftarrow \frac{1}{N} \sum_{(x,y) \in S^{(t)}} f_{\theta_t}(x) \mathbf{1}_{y=k}$
  7:     **end for**
  8:     **for** $\tau \leftarrow 0$ to $t$ **do**                ▷ Update the loss with query samples using (6).
  9:         **for** $k \leftarrow 0$ to $K$ **do**
10:             $J \leftarrow J - \sum_{(\hat{x},\hat{y}) \in Q^{(\tau)}} \log p(\hat{y} = k, \tau | \hat{x}) \mathbf{1}_{\hat{y}=k}$
11:         **end for**
12:     **end for**
13: **end for**

---

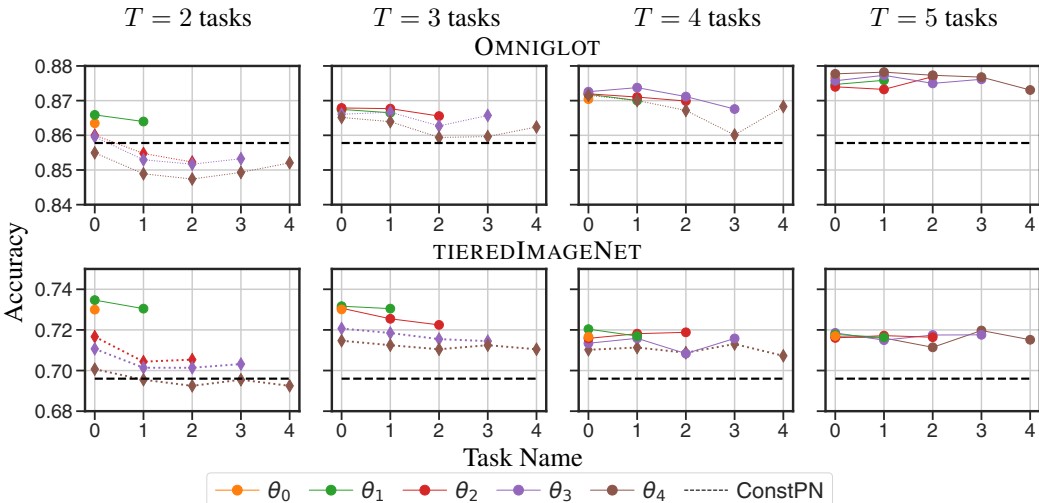

Figure 3: Task-incremental learning on OMNIGLOT and TIEREDIMAGENET. Each column represents a different IHT trained with $T = 2, 3, 4$ or $5$ tasks in total. The tasks with a bullet marker (•) correspond to the terms in the objective function (3) that are being minimized. The lines with the diamond marker (◇) are the extrapolation of the trained IHT to a larger number of tasks.

where $\ell$ is the label index and $\delta\phi = (\delta \boldsymbol{W}, \delta \boldsymbol{b}, \delta\theta)$. The lowest-order label-dependent correction to $q_\ell(x; \phi^0)$ is given simply by $s'_\ell(\cdot)(\delta \boldsymbol{W}_\ell f_{\theta^0}(x) + \delta b_\ell)$. In other words, in the lowest-order, the model only adjusts the final logits layer to adapt the pretrained embedding $f_{\theta^0}(x)$ to a new task.

For a simple softmax cross-entropy loss (between predictions $\boldsymbol{q}(x)$ and the groundtruth labels $y$), a single step of the gradient descent results in the following logits weight and bias updates:

$$\delta \boldsymbol{W}_{i,\cdot} = \frac{\gamma}{n} \sum_{(x,y) \in S} \left( \mathbf{1}_{y=k} - \frac{1}{|C|} \right) f_{\theta^0}(x), \qquad \delta b_k = \frac{\gamma}{n} \sum_{(x,y) \in S} \left( \mathbf{1}_{y=k} - \frac{1}{|C|} \right), \qquad (7)$$

where the $1/|C|$ term results from normalization in the softmax operation. Here $\gamma$ is the learning rate, $n$ is the total number of support-set samples, $|C|$ is the number of classes and $S$ is the support set. In other words, we see that the label assignment imposed by $\delta \boldsymbol{W}$ and $\delta \boldsymbol{b}$ from (7) effectively relies on computing a dot-product of $f_{\theta^0}(x)$ with "prototypes" $c_k := N^{-1} \sum_{(x,y) \in S} f_{\theta^0}(x) \mathbf{1}_{y=k}$.

## 6 EXPERIMENTS

We evaluated the model on two standard benchmark problems: 1-shot, 20-way OMNIGLOT and 5-shot, 5-way TIEREDIMAGENET. The generated weights for each task $\theta_t$ are composed of four convolutional blocks and a single dense layer. Each of the convolutional blocks consist of a $3 \times 3$

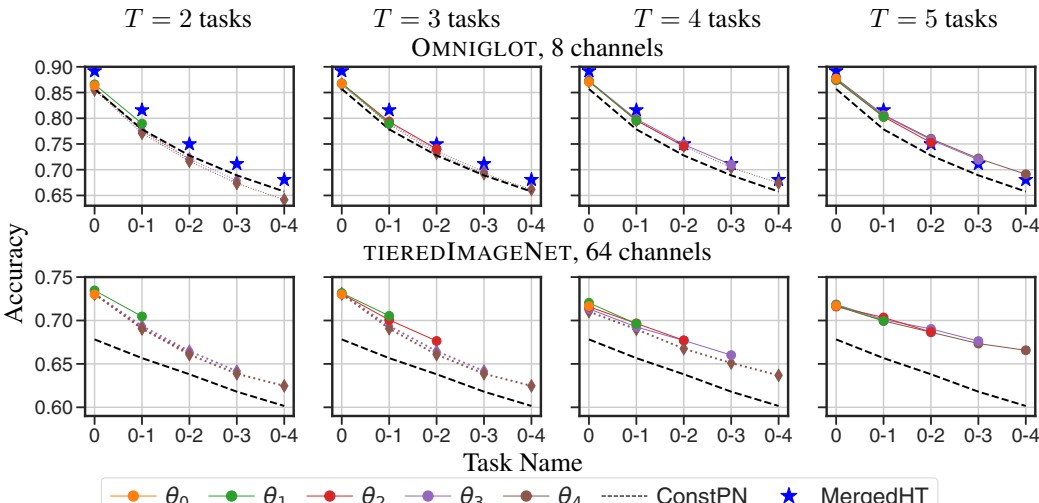

Figure 4: Class-incremental learning on OMNIGLOT and TIEREDIMAGENET. Each column represents a different IHT trained with $T = 2, 3, 4$ or $5$ tasks in total. The tasks with a bullet marker ($\bullet$) correspond to the terms in the objective function (3) that are being minimized. The lines with the diamond marker ($\diamond$) are the extrapolation of the trained IHT to a larger number of tasks.

convolutional layer, batch norm layer, ReLU activation and a $2 \times 2$ max-pooling layer. For OMNIGLOT we used 8 filters for convolutional layers and 20-dim FC layer to demonstrate how the network works on small problems, and for TIEREDIMAGENET we used 64 filters for convolutional and 40-dim for the FC layer[2] to show that the method works for large problems as well. The reported accuracy is computed for $1024$ random episodic evaluations from $\mathcal{T}_{\text{test}}$ with each episode run 16 times with different combinations of input samples.

For the HT architecture, we tried to replicate the same setup that the authors had for their original paper as closely as possible. We used a 4-layer convolutional network as a feature extractor and a 2-layer convolutional model for computing activation features. For OMNIGLOT we used a 3-layer 2-head transformer and for TIEREDIMAGENET, a simplified 1-layer transformer with 8 heads. In all our experiments we have trained the network on a single GPU for $4M$ steps with SGD with exponential LR decay over $100\,000$ steps with a decay rate of $0.97$. We noticed some stability issues with training as we increase the number of tasks and had to decrease the learning rate to adjust for it: for OMNIGLOT experiments we used a learning rate $10^{-4}$ for learning up to $4$ tasks and $5 \times 10^{-5}$ for 5 tasks. For TIEREDIMAGENET we have used the same learning rate $5 \times 10^{-6}$ for training with any $T$. As we outline above, we train the IHT models with the class-incremental objective (6), but evaluate for both task-incremental and class-incremental scenarios.

We compare our performance to two baselines. The first is a CONSTANT PROTONET (CONSTPN), which represents a vanilla Prototypical Network, as described in Snell et al. (2017). There, a universal fixed CNN network is trained on episodes from $\mathcal{T}_{\text{train}}$. This constant network can be applied to every task separately by projecting the support set as prototypes for that task and computing the prediction with respect to these prototypes. Strictly speaking, this is not a continual learning method, since it treats every task independently and has no memory of the previous tasks. For best results on this baseline we had to increase the number of classes 5 times during the training (e.g. for 20-way OMNIGLOT evaluation we have trained it with 100-way problems).

The second baseline we used specifically for the class-incremental learning is a MERGED HYPER-TRANSFORMER (MERGEDHT), where we combine all the tasks together and train a single original HT instance as a single task. This method is not a continual learning, since it has the information about all the tasks from the beginning, but it produces a solution for every class and task that we can still compare to the weights generated by the IHT.

In fig. 3 and 4 we present the main findings of our paper. In order to understand the effect of continual learning with multiple tasks, each column represents a separate run of the IHT trained on $T = 2, 3, 4$

---

[2]Note that, in contrast with cross-entropy, we do not need to have the head layer dimension to be equal to the number of predicted labels when using the Prototypical Loss.

Figure 5: UMAP projection of the IHT prototypes and the query set embeddings. The query set points are connected with the corresponding prototypes. The top plot shows the embeddings of different weights from incremental IHT training, colored according to the task. *Right plot:* UMAP projection of the CONSTPN embedding for 25 different classes from TIEREDIMAGENET

or 5 tasks in total (for training a higher $T$, please see results in the Appendix). To demonstrate the recurrence of the method, we have extended the number of tasks to 5 for the evaluation. Therefore each plot shows 5 curves that correspond to the IHT, split into two groups: bullet marker (•) for tasks that the model was trained for and diamond marker (◇) for extrapolation to more tasks. Each trained model is applied to both task-incremental (fig. 3) and class-incremental (fig. 4) setting.

For the class-incremental learning, the task name is given by two numbers indicating the range of tasks we used for evaluation (e.g. task name 0-3 corresponds to four tasks from 0 to 3).

The black constant dashed line is a baseline performance of the CONSTPN, which uses a fixed embedding and does not differentiate between the tasks. Starred blue markers represent a separate run of the HT for a particular configuration of merged tasks.

**Task-incremental learning.** We start by analysing the task-incremental learning results. For the OMNIGLOT dataset, we see no signs of catastrophic forgetting for the IHT, moreover, we observe a positive backward knowledge transfer. For a given IHT, the performance on the past tasks *improves* as more weights are being generated: e.g. for most cases the performance of $\theta_1$ (green markers) is higher than $\theta_0$ (orange markers), $\theta_2$ is higher than both $\theta_1$ and $\theta_0$ and so forth. We also noticed that as we train the IHT with more tasks, the overall performance increases: the IHT trained for $T = 5$ tasks has better results than the one trained with $T = 2$.

For the TIEREDIMAGENET the results are much better than the CONSTPN baseline, but the positive backward knowledge effect is not as pronounced as with the OMNIGLOT dataset. The performance for every training task stays roughly the same for all the generated weights. This is still a useful outcome, since it means that the model does not suffer from catastrophic forgetting.

Finally, for every trained IHT, the performance is always better than the CONSTPN baseline up to the number of tasks $T$ that it was trained for. Extrapolating the IHT to more tasks decreases the accuracy, but not significantly. Already for $T = 3$ tasks, generating 4 or 5 weights performs better than the CONSTPN.

**Class-incremental learning.** Switching to the class-incremental learning, the accuracy trends down across all the models as more and more tasks are included in the prediction. This is expected, because the size of the generated CNN does not change, but the number of overall classes that needs to be predicted is increasing. For OMNIGLOT we still observe the positive backwards transfer taking place: the IHT models trained with a higher $T$ perform better across the board. For a given model trained for a fixed $T$ the performance is comparable. This shows the preemptive property of the IHT: models trained for $T$ tasks can be still run for any number of tasks smaller than $T$ with a similar performance.

Comparing results to the baselines, the IHT has better results than the CONSTPN up to the number of tasks $T$ it was trained for, with extrapolation results improving as $T$ grows. Very surprising, for $T = 5$ the IHT was able to outperform even the MERGEDHT baseline for OMNIGLOT, which has all the information about the tasks from the beginning. We hypothesise that having more classes to classify makes the learning problem very hard for the original HT since the image embeddings are not able to learn good embeddings. This is specifically apparent for the TIEREDIMAGENET, where the numbers for the MERGEDHT are so low, they fall below the graph at 60% even for the 0-1 task.

**Multi-Domain Episode Generators.** In all of the experiments above, the support and query sets in different episodes were always sampled from the same general distribution of tasks. And even though the set of chosen image classes varied task-to-task, the overall image domain and typical

image features remained consistent across all the tasks. If different tasks could be sampled from entirely different distributions and different image domains, we would expect task-agnostic CONSTPN approach to experience accuracy degradation because it would need to identify a universal representation suited for all of the image domains at once. The IHT approach, on the other hand, could in principle adapt sample representations differently for different detected image domains.

We verify this by creating a multi-domain episode generator including tasks from a variety of image datasets: OMNIGLOT, CALTECH101, CALTECHBIRDS2011, CARS196, OXFORDFLOWERS102 and STANFORDDOGS. We measured the accuracy of CONSTPN vs. the IHT on this multi-domain generator using episodes contained two tasks with 1-shot, 5-way problems. The generated CNN model used 16 channels with the 32 channels for the final layer. Other parameters matched those used in the TIEREDIMAGENET experiments. Measured the CONSTPN accuracies were 53% for task 0, 52.8% for task 1 and 50.8% for combined tasks. For IHT accuracies were 56.2% for task 0, 55.2% for task 1 and 53.8% for combined tasks. The accuracy gap of nearly 3% between these two methods (higher than that for OMNIGLOT and TIEREDIMAGENET) suggests that the IHT is better at adapting to a multi-domain task distribution, but additional experiments are needed to confirm that adding new image domains hurts the CONSTPN performance more than it does the IHT.

**Analysis of prototypical loss.** We want to take a closer look at prototypes and query set embeddings for generated weights. For a given IHT trained for $T = 5$ on TIEREDIMAGENET 5-way, 5-shot problem, we picked a random test of 5 tasks and ran them through the model, producing weights $\theta_0$ to $\theta_4$ along with prototypes for each class for every task. We then computed the logits of the query sets consisting of 20 samples per class for every task. The resulting 40-dim embeddings for prototypes and query sets are concatenated and projected to 2D using UMAP (fig. 5). Notice that the embeddings of the tasks are well separated in the logits space, which explains why the model performs well for both task- and class-incremental learning. Normalizing the softmax over the classes from the same tasks or across the classes for all the tasks makes little difference when the tasks

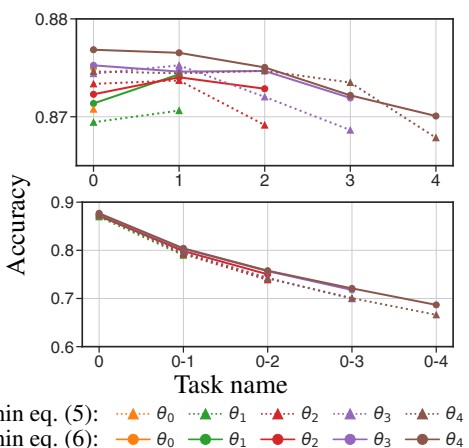

Figure 6: IHT trained using task-incremental objective (5) vs. class-incremental objective (6).

are so far away from each other. On the right of fig. 5 we show the projection of the CONSTPN embedding of the same 25 classes. The CONSTPN does not make a distinction between tasks and treats every class separately. The fact that we see 3 clusters emerge has to do purely with the semantics of the chosen clusters and the way the CONSTPN decides to group them. This also sheds light on why the IHT performs better than the CONSTPN, since it separates the tasks first before separating the classes within each task.

Finally, in fig. 6 we show the accuracy of two different models trained with task-incremental (5) and class-incremental objectives (6). The performance of both models on task-incremental problems are comparable, while for class-incremental the model trained for that objective is better.

## 7 CONCLUSIONS

The presented INCREMENTAL HYPERTRANSFORMER model has several desirable properties. On one hand, as an efficient few-shot learner, it is able to generate CNN weights on the fly with no training required from a small set of labeled examples. On the other hand, as a continual learner, it is able to update the weights with information from a new task by piping them recursively through a new iteration of the HT. We showed empirically that the learning happens without the catastrophic forgetting and, in some cases, even with a positive backward transfer. By modifying the loss from cross-entropy to prototypes we were able to define a learning procedure that optimizes the location of the prototypes of every class for each task. A single trained IHT model works in both task- and class-incremental regimes.

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

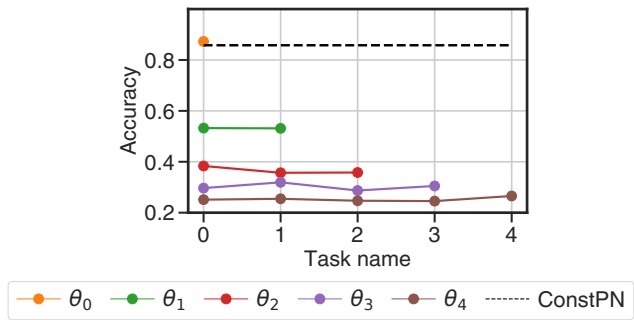

Figure 7: The accuracy of the HT trained for $T = 5$ using cross-entropy loss. While the accuracy of the first weight $\theta_0$ is high and is better than the accuracy of the CONSTPN embedding, it drops dramatically when more tasks are added, because of collisions between the same classes for different tasks in the cross-entropy loss.

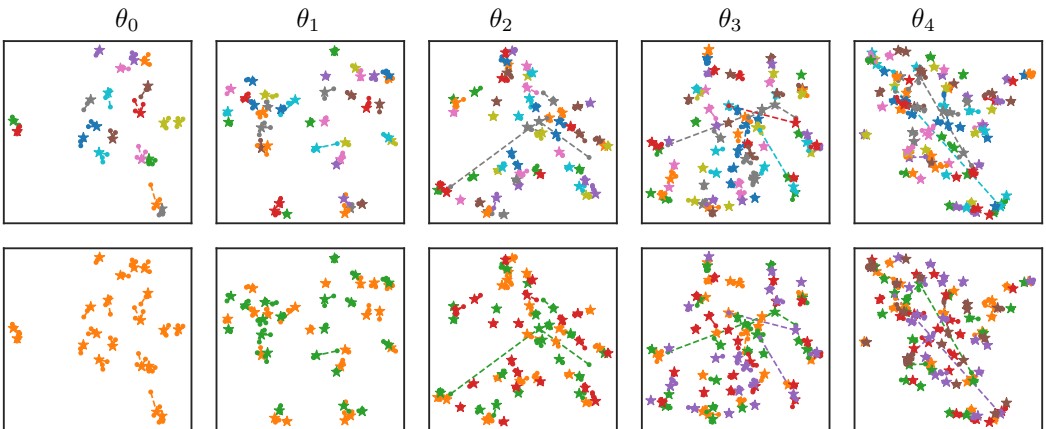

Figure 8: 2-dim UMAP projection from 20-dim embedding of the support set (prototypes) and query set for different weights from incremental HT training. The query set points are connected with their corresponding prototypes with a line. The top and bottom plots are identical, but colored according to both their class and task (*top* plot), or just according to their task (*bottom* plot). Notice that the prototypes from the earlier tasks remain the same for the later task, however since UMAP projection is non-parametric and have to be re-run for every new $\theta_k$, the 2-dim projection of the prototypes is not the same. We tried our best to align the embedding using the Procrustes alignment.

## A  ADDITIONAL FIGURES

### A.1  LEARNING WITH CROSS-ENTROPY LOSS

Figure 7 shows an attempt to do learn incrementally from multiple tasks using a HT with a cross-entropy loss. Since we do not increase the size of the last layer's embedding, we can predict only class and not task, which corresponds to the domain-incremental learning setup. Moreover, the same class from different tasks are mapped to the same location in the embedding space, thus creating collisions when more tasks are added. This is why the accuracy drops very dramatically when the number of tasks increase. On the other hand, CONSTPN is more flexible, since the prototypes for each tasks are computed from the support set of a given task and do not have to be fixed to the one-hot vector, as it done for the cross-entropy.

### A.2  UMAP PLOT OF THE MULTI-TASK PROTOTYPES

UMAP embedding for the OMNIGLOT embedding using ProtoNet (fig. 8) looks different from similar embedding projection of TIEREDIMAGENET. In particular, it looks like the embeddings

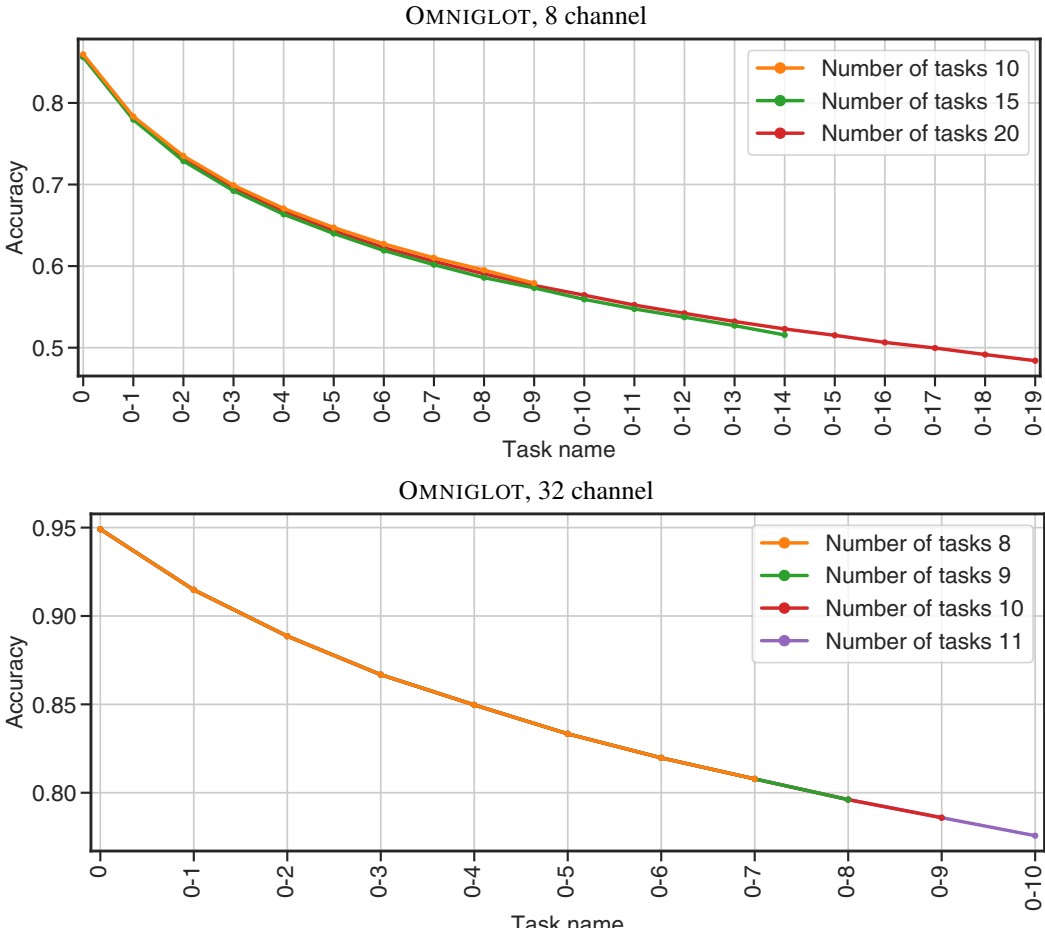

Figure 9: OMNIGLOT with 8 or 32 channels trained with a different number of tasks $T$.

from different tasks are overlapping, in contrast to the TIEREDIMAGENET embedding that are separated. We think this is because the classes in OMNIGLOT are much more closely connected than for TIEREDIMAGENET. Curiously, even though the classes between the tasks are overlapping, the final accuracy is still in the high $90\%$ and barely degrades when more tasks are added.

### A.3 LEARNING WITH MORE TASKS

Although, we mainly analysed the results of IHT for up to 5 tasks, we can run the IHT for much more than that. Fig. 9 shows the IHT run for a different number of total tasks $T$ for the OMNIGLOT dataset with 8 and 32 channels convolutional layers. Similar to the results in the main paper, the nearly overlapping curves mean that the model trained for $T$ tasks can generalize and run for up to that many tasks with the same accuracy as the model trained for a smaller number of tasks.

### A.4 INCREMENTAL HYPERTRANSFORMER VS MERGEDHT FOR TIEREDIMAGENET

Fig. 10 shows the zoomed out view of the same figure presented in the main paper (fig. 4). Here, one can clearly see how much worse the MERGEDHT is compared to the IHT.

### A.5 ADDITIONAL FIGURES FOR OMNIGLOT TASK- AND CLASS-INCREMENTAL LEARNING

Fig. 11, 12, 13 and 14 show additional experiments with OMNIGLOT with different number of channels in the CNN.

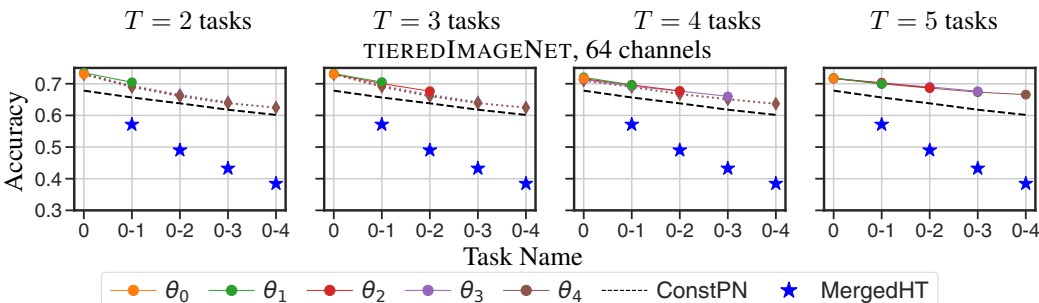

Figure 10: Zoomed out view of fig. 4 so that the results of the MERGEDHT is visible.

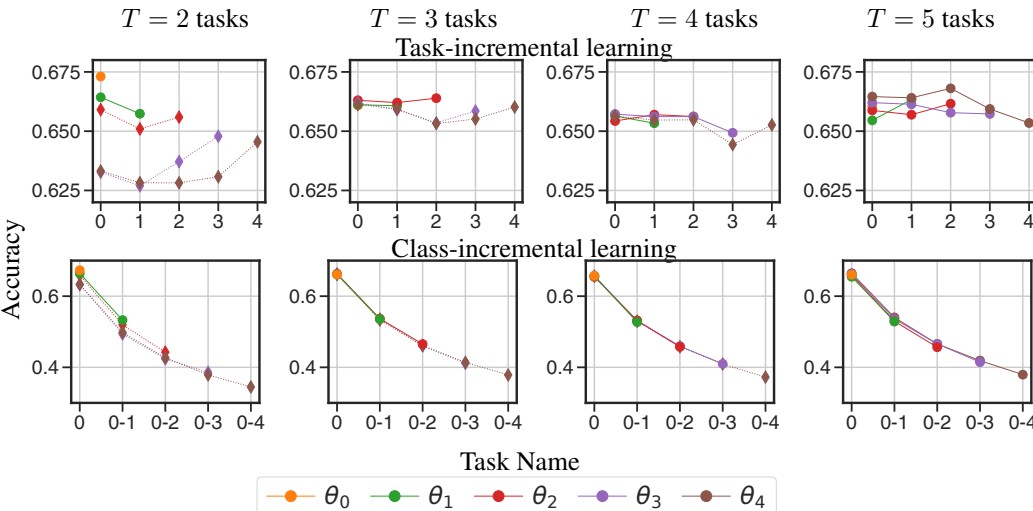

Figure 11: Task- and class-incremental learning on OMNIGLOT with 4-channels convolutions.

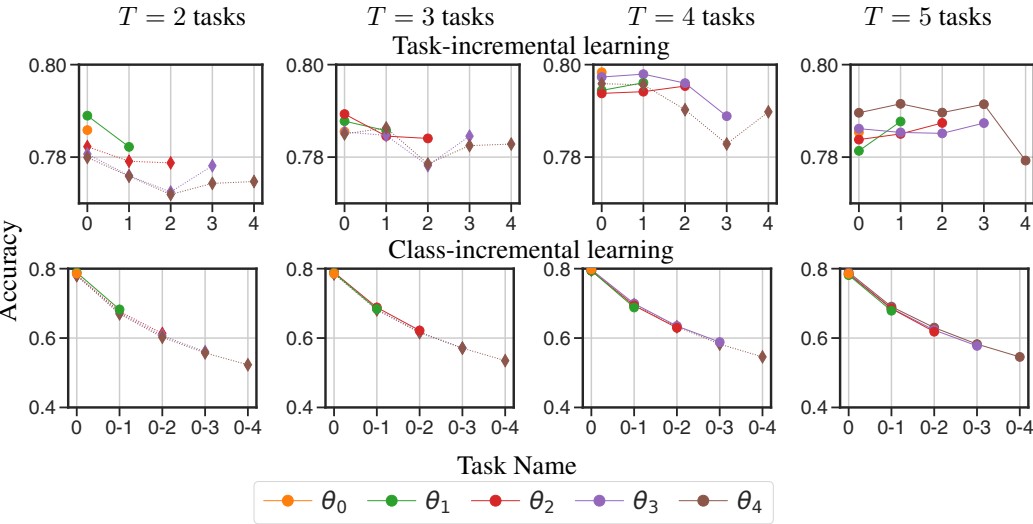

Figure 12: Task- and class-incremental learning on OMNIGLOT with 6-channels convolutions.

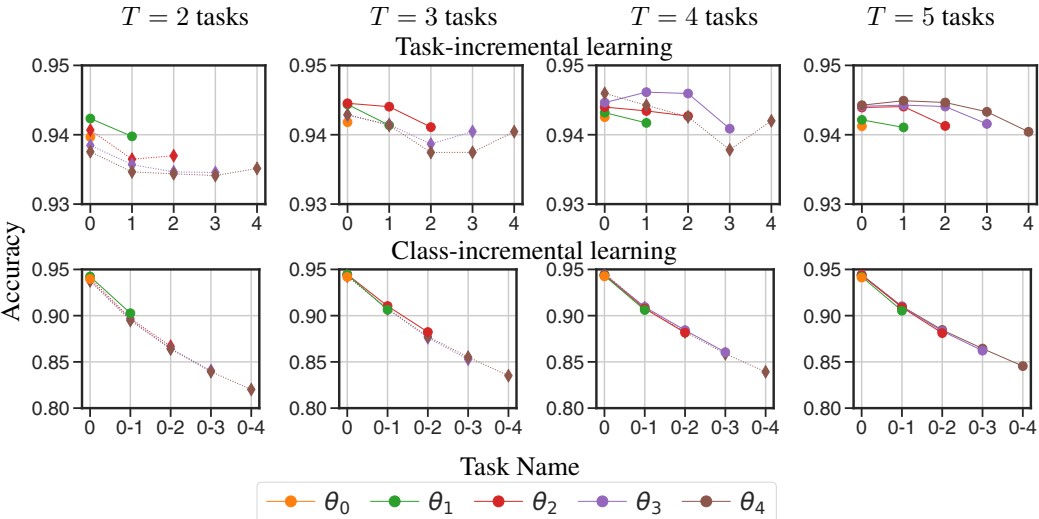

Figure 13: Task- and class-incremental learning on OMNIGLOT with 16-channels convolutions.

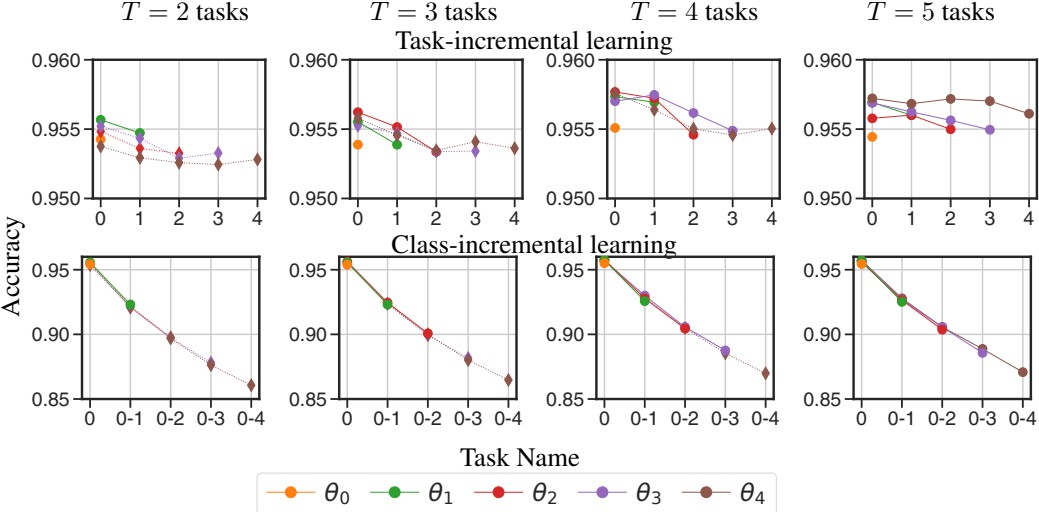

Figure 14: Task- and class-incremental learning on OMNIGLOT with 32-channels convolutions.

