# OpenReview forum: "Few-Shot Incremental Learning Using HyperTransformers"
_ICLR.cc/2023/Conference — Submitted to ICLR 2023_

### Official Review · Reviewer_hN2L · 2022-10-22

**Confidence:** 3
**Correctness:** 4
**Technical Novelty And Significance:** 3
**Empirical Novelty And Significance:** 3
**Recommendation:** 5

**Clarity, Quality, Novelty And Reproducibility:**

I think the definition of the core problem in this paper is missing, and I expect more explanation about why the proposed method can alleviate forgetting.


**Strength And Weaknesses:**

Strength:
- The architectural innovation seem effective and are easily interpretable when thinking about utilizing the learned knowledge.
- The prototypical loss is orthogonal to the architecture design.
- I think even general classification task can benefit from the proposed architecture design and loss design.

Weaknesses:
- I think it is better to give a clear definition of few-shot class-incremental-learning
- In my opinion, the re-use of old weights can help the convergence of the new task because the old weights can be treated as an initialization of the new model, but how does this mechanism alleviate the catastrophic forgetting? Does author use the replay buffers? It seems that we have to store all the query sets of each seen tasks.
- The proposed prototypical loss is similar to the Nearest-Mean-of-Exemplars Classification in iCaRL[1]. More discussion is expected.
- The experiments lack comparison with other continual learning baselines.


[1] Rebuffi S A, Kolesnikov A, Sperl G, et al. icarl: Incremental classifier and representation learning. CVPR 2017.

**Summary Of The Paper:**

This paper extended the HyperTransformer and proposed an Incremental HyperTransformer(IHT). The proposed IHT re-used the old weights as an input to generate the new weights for the incoming task in the continual sequence. This mechanism encourages the new model to utilize the knowledge in the old model. Experiments on class-incremental setting and task-incremental setting demonstrate the effectiveness of the proposed method.

**Summary Of The Review:**

Based on the comments in the Main Review part, I tend to reject this paper. The main reason is the lack of explanation and experiments of why the proposed method can catastrophic forgetting.

---

> ### Author Response · Authors · 2022-11-11
> **Answer to the Reviewer hN2L**
>
> We thank the reviewer for the time they took to review the manuscript. We are glad that they found the design effective and easily interpretable.
>
> > I think it is better to give a clear definition of few-shot class-incremental-learning
>
> Thanks for the suggestion! We do speak about class- and task-incremental learning in the introduction: “The IHT works in two different continual learning scenarios: task-incremental learning (predict class attribute using the task information) and class-incremental learning (predict both class and task attributes).” However, we agree that we should be more precise when we define this: few-shot class-incremental learning learns tasks sequentially from a few labeled examples per task, such that the final weights $\theta_t$ can predict both class and task attributes of the samples from all the classes of all the tasks $\tau=0\dots t$ that it had seen during learning.
>
> > In my opinion, the re-use of old weights can help the convergence of the new task because the old weights can be treated as an initialization of the new model, but how does this mechanism alleviate the catastrophic forgetting? Does author use the replay buffers? It seems that we have to store all the query sets of each seen tasks.
>
> That is a great question. The query sets are used _only_ during the training to compute the loss value and backpropagate the results to update the HT weights $\psi$. Once the CNN weights $\theta_t$ are produced, we evaluate the loss on each of the prior task $\tau=0\dots t$ (see eq. (2)). This is exactly how IHT alleviates the catastrophic forgetting: CNN weights $\theta_t$ are minimized on the query set of every task seen before.
>
> During the evaluation, the HT weights $\psi$ are already trained and fixed. For each of the learned tasks, _only_ the support set is passed through $\psi$ in order to generate the weights $\theta$. We do compute the accuracy using the query set, but this is just a ground truth to measure against.
>
> > The proposed prototypical loss is similar to the Nearest-Mean-of-Exemplars Classification in iCaRL[1]. More discussion is expected.
>
> Thanks for the suggestion. We will be happy to include in the paper the discussion on other alternatives to the prototypical loss, including the suggested iCaRL algorithm. Indeed, since 2017 when the ProtNet paper was published, there were a lot of modifications and improvements to the method (e.g. [1,2]). However, we have found that the a simple vanilla prototypical loss serves our need to create trainable "anchor" points for classes and tasks from the support set. We believe that simplicity of the approach allows to focus on what is important: trainable class/task representations in ProtoLoss instead of the fixed location of the Cross Entropy enable to learn continually for both class- and task-incremental settings.
>
> [1] Oreshkin, B., Rodríguez López, P., & Lacoste, A. TADAM: Task dependent adaptive metric for improved few-shot learning. In NeurIPS, 2018.
>
> [2] Zhang, C., Cai, Y., Lin, G., & Shen, C. DeepEMD: Few-shot image classification with differentiable earth mover's distance and structured classifiers. In CVPR, 2020.
>
> > The experiments lack comparison with other continual learning baselines.
>
> What we propose is not just a continual learning method, but an algorithm for a combination of few-shot and continual learning. We have compared our method to the baselines most closely related to our methods ($ConstPN$ and $MergedHT$) and added a discussion about other incremental few-shot learning methods to the Related work section.
>
> Many of the existent incremental few-shot methods define the problem differently than we do: they train a classifier model on many base classes with _abundant_ examples and then adapt this model to a new task with few labeled data points while not forgetting base classes. In contrast, our model does not have a distinction between the base classes and novel inference few-shot leaning training. We learn all the tasks in an episodic fashion by sampling at random from a set of tasks $T_{train}$ recursively one by one. For the inference, we sample from a different set of tasks $T_{test}$ to check the performance of the trained IHT.
>
> We were not able to find other papers that have a similar formulation of recurrent continual learning from the few-shot data. We would be happy to add references and comparison if the reviewer suggests any that we might have missed.

---

> > ### Author Response · Authors · 2022-11-24
> > **Answer to the Reviewer hN2L**
> >
> > Dear Reviewer hN2L,
> >
> > It looks like the main reason why you are leaning towards the rejection of our paper is lack of explanation why the proposed method is able to alleviate the catastrophic forgetting. We have tried to clarify this for you in the answer above. Indeed, we think of this property as one of the fundamental features of our method: the IHT learns to encode weights $\theta_{t}$ such that not only they perform well as a CNN, but also that they retain the information about the past tasks when learning about the following task $t+1$. This is different from other continual learning methods that use replay buffers or explicit weight regularization methods (e.g. using Fisher Information as in EWC method). In our method, the HyperTransformer itself is trained to figure out how to encode the weights such that the information is not lost.
> >
> > We hope that this answer helps explain a bit better how our method works. We are happy to continue the discussion and provide any additional information as needed.

---

### Official Review · Reviewer_XnH3 · 2022-10-23

**Confidence:** 3
**Correctness:** 3
**Technical Novelty And Significance:** 2
**Empirical Novelty And Significance:** 2
**Recommendation:** 3

**Clarity, Quality, Novelty And Reproducibility:**

The paper is interesting and in some sense presents valuable novelty. Unfortunately, the presentation and evaluation of the model are not enough for publication.

**Strength And Weaknesses:**

1. The paper is not well written. For me, it is not trivial to extract information about the setting. What authors mean by continual learning in few-shot learning.

FEW-SHOT INCREMENTAL LEARNING is well described in literature and authors should concentrate on such a problem, not few-shot or continual learning.

2. The paper should contain a section that explains what authors understand by continual learning mixed with few-shot learning. How exactly looks FEW-SHOT INCREMENTAL LEARNING in the paper?

3. Chapter 3 HYPERTRANSFORMER ARCHITECTURE starts with the formulation of few-shot learning. But section 4 INCREMENTAL HYPERTRANSFORMER directly starts with a model, without a specification of how the few-shot learning task was generalized.

4. Experimental section is not well done. First of all, it is not clear what the training looks like and how the evaluation is computed.

5. Authors do not specify numerical measures for evaluations

6. Authors evaluate the method only by plots. It is difficult for future investigators to work whit such papers. The charts are nice the authors should give exact numbers in the appendix.


**Summary Of The Paper:**

In the paper, the authors use HyperTransformer (HT) in a new setting that mixes continual learning and few-shot learning. The authors propose to reuse these generated weights as input to the HT for the next task.

**Summary Of The Review:**

The paper is interesting but the presentation and evaluation of the model are not enough for publication.

---

> ### Author Response · Authors · 2022-11-11
> **Answer to the Reviewer XnH3**
>
> > The paper is not well written. For me, it is not trivial to extract information about the setting. What authors mean by continual learning in few-shot learning.
>
> We thank the reviewer for their time. It is apparent that the clarity was the main concern of the reviewer. We will be happy to make an additional pass on the paper and improve the clarity and readability. We also hope that our answer below clarifies the concerns that the reviewer has. We will be happy to continue the discussion and address any additional questions that the review might have.
>
> > FEW-SHOT INCREMENTAL LEARNING is well described in literature and authors should concentrate on such a problem, not few-shot or continual learning. The paper should contain a section that explains what authors understand by continual learning mixed with few-shot learning. How exactly looks FEW-SHOT INCREMENTAL LEARNING in the paper?
>
> Section 4 describes the problem that we are trying to solve:
> "We consider a continual stream of tasks sampled sequentially at random without replacement. [...] For each task $t$, the IHT receives the support set of that task $S(t)$ as well as the previous weights $θ_{t−1}$ and generates the weights.[...] The IHT sequentially generates a series of weights {$\theta_\tau$}$^{t}_{\tau=0}$, each of which are suited for all the tasks coming before it."
>
> In the related work section of our paper we discuss how our approach differs from other Incremental few-shot learning papers (page 3). We will be happy to address specific questions from the reviewer on why this section seems not complete to them.
>
> > Chapter 3 HYPERTRANSFORMER ARCHITECTURE starts with the formulation of few-shot learning. But section 4 INCREMENTAL HYPERTRANSFORMER directly starts with a model, without a specification of how the few-shot learning task was generalized.
>
> The few-show learning task is not generalized, it actually stays the same. Instead we consider a stream of few-shot learning tasks coming sequentially, thus creating a continual few-shot learning problem. In our model, the generated weights from a given task also serve as the input to the following few-shot learning problem on a different task. We demonstrate that the generated weights are suited for all the tasks that IHT has seen before without the catastrophic forgetting or, in some cases, even with a positive backward transfer.
>
> > Experimental section is not well done. First of all, it is not clear what the training looks like and how the evaluation is computed.
>
> Section 6 describes the experimental setup of the paper. Specifically, the first two paragraphs outline the dataset and the model setup that we have used. The training is done by passing $T$ tasks selected at random from a set of tasks $T_{train}$ through the proposed IHT method. After the loss is computed, we use SGD to propagate the gradients back and compute the update of the HT weights. We repeat this for a different set of tasks from $T_{train}$ till convergence (for a max of $4M$ iterations). After that, HT weights are fixed and the evaluation step begins. There, we sample $T$ tasks from a set $T_{test}$ and pass those tasks through a trained HT model to generate necessary weights for which we compute the accuracy of this episode. We repeat the procedure 1024 times for different choices of tasks and samples within each task and average the results to produce the plots that you see at Figs. 3 and 4.
>
> > Authors do not specify numerical measures for evaluations
>
> We would be curious to learn what exactly the reviewer means by the "numerical measures". We measure the performance of the few-shot learning by reporting the mean accuracy of each few-shot learning task, as it is done throughout the few-shot learning literature.
>
> > Authors evaluate the method only by plots. It is difficult for future investigators to work whit such papers. The charts are nice the authors should give exact numbers in the appendix.
>
> Thanks! This definitely would be helpful for results comparison. We would be happy to add such a table in the appendix.

---

> > ### Author Response · Authors · 2022-11-24
> > **Answer to the Reviewer XnH3**
> >
> > Dear Reviewer XnH3,
> >
> > In the answer above, we tried to clarify the points raised in the review. We hope our response was helpful. We would be grateful if the reviewer could acknowledge our answer and we are happy to continue the discussion if needed.

---

> > > ### Comment · Reviewer_XnH3 · 2022-11-28
> > > **Final score**
> > >
> > > Thank you for the answers, but I still am not convinced. In my opinion, the article is not ready for publication. I maintain my original score.

---

### Official Review · Reviewer_UZsQ · 2022-10-25

**Confidence:** 4
**Correctness:** 3
**Technical Novelty And Significance:** 2
**Empirical Novelty And Significance:** 2
**Recommendation:** 3

**Clarity, Quality, Novelty And Reproducibility:**

Clarity is good.
Quality is poor due to the limited novelty.
Not sure about the reproducibility. No code was given.

**Strength And Weaknesses:**

Strength:

It is a good baseline method for using the weight generation model in class-incremental and task-incremental learning.

Weakness:

Its applications of the existing weight generation model and the typical prototypical loss are straightforward. Its overall idea is too similar to many related few-shot learning works (which however not being mentioned in the related work section). Please check some of them in [1,2,3]. It does not explicitly solve the forgetting problems in incremental learning, but simply relies on the performance of the adopted hypernetwork. I can take this work as a good baseline but without clear novelty over the baseline.

[1] Li et al. CVPR 2022. Sylph: A Hypernetwork Framework for Incremental Few-shot Object Detection. It introduced a few-shot hyper network to generate weights and biases for each few-shot detection task.

[2] Przewięźlikowski et al. arXiv 2022. HyperMAML: Few-Shot Adaptation of Deep Models with Hypernetworks. It introduced HyperMAML — an approach to the Few-Shot learning problem by aggregating information from the support set and directly producing weights updates.

[3] Sendera et al. ArXiv 2022. HyperShot: Few-Shot Learning by Kernel HyperNetworks. It fused kernels and hyper network paradigm. Its hypernetwork takes the support set as input to predict the classifier weights to do few-shot task adaption.

**Summary Of The Paper:**

This paper makes a few contributions: using a newly published weight generation model HyperTransformer to generate model weights for incremental task/class learning; modifying the classic SCE loss with an existing prototype loss (from a few-shot learning paper).

**Summary Of The Review:**

It is a good baseline paper based on existing techniques (or losses), but does not contribute a novel idea or solution to class-incremental learning or task-incremental learning explicitly. The paper is clearly written and easy to follow.

---

> ### Author Response · Authors · 2022-11-11
> **Answer to the Reviewer UZsQ**
>
> We thank the reviewer for their time. We would like to take this opportunity to address their criticism below in the hopes to change their recommendation towards our paper. We will be happy to continue the discussion with the reviewer if they have any additional comments or suggestions.
>
> > Its applications of the existing weight generation model and the typical prototypical loss are straightforward. It is a good baseline paper based on existing techniques (or losses), but does not contribute a novel idea or solution to class-incremental learning or task-incremental learning explicitly.
>
> We would like to respectfully disagree with the reviewer. The main idea of the paper – to generate the weights for the few-shot learning problem _and_ re-use the same weights as a representation of the past task is significantly different from a typical continual learning method (such as using a replay buffer, making per-task model adjustments or using a regularization loss). IHT is trained directly to update the weights with new information about the new task without forgetting the previous ones. Our experimental results confirm that this idea is effective: we were able to demonstrate that our system is not suffering from the catastrophic forgetting and, in some cases, even show positive backward transfer.
>
> We think that the papers mentioned by the reviewers are sufficiently different from our proposed model for the reasons we outline below (e.g. only one of them is actually solving a problem that is both _incremental_  and _few-shot_). We indeed missed them in our literature review (looks like they all were released within the last 6 months) and we will be happy to incorporate them into our literature review.
>
> > [1] Li et al. CVPR 2022. Sylph: A Hypernetwork Framework for Incremental Few-shot Object Detection. It introduced a few-shot hyper network to generate weights and biases for each few-shot detection task.
>
> The problem that this paper studies consists of (a) learning from a set of base classes with _abundant_ data and then (b) adapting the model to a novel few-shot object-detection problem without forgetting the base classes. Notably, we notice few properties of their set up:
>
> - Their base learning is learned on the abundant labeled data.
> - The adaptation problem (b) is designed to be different from the learning problem (a).
> - The adaptation is happening on top of the base classes. It is not possible to keep learning new classes on top of the existing adaptation.
>
> Instead, we propose to solve a different problem. We don’t have a distinction between the base training and novel inference training. We learn in an episodic fashion by sampling tasks at random from a set of tasks $T_{train}$ and learning them one by one using IHT. For the inference, we sample from a completely different set of tasks $T_{test}$ to check the performance of the trained IHT.
>
> In addition, we train a single universal model – a HyperTransformer – that is able to update the weights _recursively_ with information from the support set of a novel task without forgetting about previously learned tasks.
>
> > [2] Przewięźlikowski et al. arXiv 2022. HyperMAML: Few-Shot Adaptation of Deep Models with Hypernetworks. It introduced HyperMAML — an approach to the Few-Shot learning problem by aggregating information from the support set and directly producing weights updates.
>
> This paper proposes a modification to a few-shot learning MAML algorithm, where the adaptation to a support set is given by a hypernetwork instead of using the gradient descent. The paper doesn’t mention continual or incremental learning and adapting their setting to a few-shot incremental problem is not trivial.
>
> > [3] Sendera et al. ArXiv 2022. HyperShot: Few-Shot Learning by Kernel HyperNetworks. It fused kernels and hyper network paradigm. Its hypernetwork takes the support set as input to predict the classifier weights to do few-shot task adaption.
>
> Similar to the paper above, this paper addresses only the few-shot learning part by predicting the parameters of the kernel representation of the support set  that is used to classify the query data. The paper doesn’t mention continual learning and it is not obvious how this method can be used for the continual few-shot learning.

---

> > ### Author Response · Authors · 2022-11-24
> > **Answer to the Reviewer UZsQ**
> >
> > Dear Reviewer UZsQ,
> >
> > In the answer above, we tried to clarify the points raised in the review. We hope our response was helpful. We would also like to provide an additional clarification on the way our method deals with catastrophic forgetting:
> >
> > > "[IHT] does not explicitly solve the forgetting problems in incremental learning, but simply relies on the performance of the adopted hypernetwork."
> >
> > The only methods that we know that _explicitly_ solve the forgetting problem rely on weight masking conditioning on the task_id. Neither replay buffer methods, nor regularization methods such as EWC solve for forgetting _explicitly_.  Indeed, the performance of the hypernetwork allows to generate the weights that prevent catastrophic forgetting. We consider this a strength and a unique properly of our method and our experiments demonstrate the effectiveness of this approach.
> >
> > We would be grateful if the reviewer could acknowledge our answer and we are happy to continue the discussion if needed.

---

### Official Review · Reviewer_qvBd · 2022-10-30

**Confidence:** 4
**Correctness:** 3
**Technical Novelty And Significance:** 4
**Empirical Novelty And Significance:** 3
**Recommendation:** 8

**Clarity, Quality, Novelty And Reproducibility:**

In my opinion, the proposed method is novel. The paper is well written and easy to follow. No code is provided therefore I can't say much about the reproducibility of the work.

**Strength And Weaknesses:**

### Strengths
- The approach might seem like simple extension of hypertransformers, but to make it work is not easy. Thus I believe it's novel and substantial contribution.
- The paper is fairly well-written and easy to follow.
- The results are convincing but could have been better. I expand on this in the following section.

### Weaknesses
- I would have appreciated a slightly harder task for tiered imagenet i.e. 20 way, 5 shot.
- From task incremental learning (fig 3), it seems that adding more tasks does lead to some forgetting as performance for $\theta_{0}$ and $\theta_{1}$ is consistently higher than $\theta_{3}$ and onwards (esp. for tiered imageNet). Could authors explain why is this the case?
- I would have appreciated if the authors had considered some external few-shot continual learning baselines such as FSCL from [Wang et al. 2021b].
- Instead of freezing the prototypes, could they be updated with momentum update as it was done in momentum contrastive learning [1]?
- What  does the x-axis of individual plots in figure 3 refers to?
- Why is there is no MergedHT baseline for tiered imageNet (fig4) and task incremental learning (fig3)?
- Since you use task embeddings as a state (like in recurrent neural networks). Do these embeddings get updated in between tasks by some other mechanism? For context, I am trying to understand if there could be an issue like vanishing gradients with the proposed recurrent neural network like approach?

**Summary Of The Paper:**

The paper proposes a few-shot continual learning approach that uses a hypertransformer to infer the task-specific CNN weights. These weights are then used as inputs to the hypertransformer thus serving as a state for it. The whole architecture resemble a recurrent neural network but with added feature of assimilating few-shot learning. I liked the idea of extending hypertransformers to continual learning.

**Summary Of The Review:**

The paper proposes a new way to assimilate continual learning in hypertransformers architecture. I liked the approach, but have few concerns regarding the evaluations. I have pointed them out in the Weaknesses and would appreciate if the authors could address them.

---

> ### Author Response · Authors · 2022-11-11
> **Answer to the Reviewer qvBd #1**
>
> We thank the reviewer for their time and for providing useful feedback to our paper. We are glad that the reviewer has found that the paper is well-written and constitutes a substantial contribution with convincing results.  We hope that our comments below answer the concerns that the reviewer had and we are happy to provide more details if needed.
>
> > I would have appreciated a slightly harder task for tiered imagenet i.e. 20 way, 5 shot.
>
> Thanks for the suggestion. In a way, we already have this results in the paper. _20-way, 5-shot_ problem corresponds to our $MergedHT$ baseline, where 4 tasks each of _5-way, 5-shots_ are combined together as a single problem. The results were quite bad and fell out of the chart in Fig. 4 (we have included Fig. 10 in the Appendix that specifically shows a zoomed out view of Fig.4 so that the $MergedHT$ is visible). It turns out that the performance of the HyperTransformer drops quite rapidly when the number of classes (way) is increasing (at least for a complex problem such as tieredimageNet). Very curiously, the results of the tieredImageNet _20-way 5-show_ can also be achieved by using a proposed IHT as an incremental learning problem of 4 tasks _20-way, 5-shot_ each. In this case, the performance is much better and can be visible in the same Fig. 4 and 10. Essentially, if the number of classes in your task is large, our recommendation is to run IHT instead of the original HT, _even if your data is not presented continually_. We will make this point more prominent in the paper.
>
> > From task incremental learning (fig 3), it seems that adding more tasks does lead to some forgetting as performance for $\theta_0$ and $\theta_1$ is consistently higher than $\theta_3$ and onwards (esp. for tiered imageNet). Could authors explain why is this the case?
>
> In Fig.3 the performance of $\theta_0$ and $\theta_1$ is higher only for the cases where the number of trained tasks $T$ is 2 or 3. However, for $T=2$ $\theta_0$ and $\theta_1$ are the only ones that are being optimized and $\theta_2, \theta_3$ and others (essentially all the weights marked with a diamond marker) are generated during the meta-test just to show that our system is indeed recurrent and generalizes beyond the number of tasks $T$ it was trained on. It is expected that the extrapolation beyond $T$ would not work as good as compared to the weights that we trained for. If one compares only the trained weights to each other (solid lines with a bullet marker), forgetting doesn’t seem to happen.
>
> > I would have appreciated if the authors had considered some external few-shot continual learning baselines such as FSCL from [Wang et al. 2021b].
>
> Thanks for pointing it out. The setting of that paper is different from ours. They train a classifier model on _base_ classes with abundant examples and then adapt this model to a new task with few labeled data points while not forgetting base classes. The key difference here is training on _abundant_ examples from the base classes and a _separate_ mechanism to extend the result to a novel few-shot problem that is done only _once_ at the inference time. In contrast, our model does not have a distinction between the base classes training with a lot of labels and novel inference few-shot leaning training. We learn all the tasks in an episodic fashion by sampling at random from a set of tasks $T_{train}$ and learning them recursively one by one. For the inference, we sample from a different set of tasks $T_{test}$ to check the performance of the trained IHT.
>
> > Instead of freezing the prototypes, could they be updated with momentum update as it was done in momentum contrastive learning [1]?
>
> This is an interesting suggestion and indeed in our paper we went with the easiest approach of simply freezing the prototypes. During the initial experiments, we have found out that, surprisingly, freezing the prototypes works exactly the same as updating the prototypes for new generated weights, which is pretty much as good as you can get. We think that for harder problems, for example where tasks’ semantic changes more dramatically from task to task (and subsequently the generated weights change a lot), simply freezing the prototypes might not work as well and additional techniques such as momentum contrastive learning would be helpful.

---

> > ### Author Response · Authors · 2022-11-11
> > **Answer to the Reviewer qvBd #2**
> >
> > > What does the x-axis of individual plots in figure 3 refers to?
> >
> > x-axis in Fig.3 refers to the Task Name. In the beginning of the training, IHT is presented with Task 0, generates weight $\theta_0$ and is evaluated on Task 0 (orange dot on the first column “0”). IHT then takes $\theta_0$ and Task 1, generates weights $\theta_1$ and is evaluated on Task 0 and Task 1 (green dots on the column “0” and ”1”). Then $\theta_2$ is generated from Task 2 and $\theta_1$ and evaluated on Tasks 0, 1 and 2. And so forth…
> >
> > For the Fig.4, the difference is that the IHT is evaluated on the _combined_ task (i.e. predicting both class_id and task_id), e.g. $theta_1$ is evaluated on Task “0” and Task “0-1” (which is a combined prediction of tasks 0 and 1 together).
> >
> > > Why is there is no MergedHT baseline for tiered imageNet (fig4) and task incremental learning (fig3)?
> >
> > As we explain above, $MergedHT$ for tieredImageNet from Fig.4 is actually there, but it is so bad it fell out of the chart. We present this full zoomed out result in Fig.10.
> > For task-incremental learning (Fig.3), there is no $MergedHT$ since every task is evaluated separately and no tasks are actually merged.
> >
> > > Since you use task embeddings as a state (like in recurrent neural networks). Do these embeddings get updated in between tasks by some other mechanism? For context, I am trying to understand if there could be an issue like vanishing gradients with the proposed recurrent neural network like approach?
> >
> > This is a very interesting observation that we also have thought about. In many ways, the vanishing gradients is not a problem in our setup, since we evaluate the loss for _every newly generated $\theta_t$_: $\theta_0$ is evaluated on Task 0,  $\theta_1$ is evaluated on Task 0 and Task 1 and so forth. This way, there are loss terms for every task (see also Fig.2) and the vanishing gradient is not a problem.
> >
> > However, we did try two experiments to understand this effect better:
> > - We have tried to minimize only the last weight. For example, for 3 tasks, we only minimize the performance of the $\theta_2$ on Tasks 0,1,2 and ignore the performance of $\theta_0$ and $\theta_1$. This can be useful for the cases when the practitioner knows exactly how many tasks are there and is not interested in the performance of the intermediate weights. In this scenario, probably because of the vanishing gradients problem, the results were not as good as the setup that we used in the paper: unsurprisingly, the performance of the $\theta_0$ and $\theta_1$ have dropped, but the performance of $\theta_2$ never reached numbers as high as the one that we show in the paper.
> > - In order to study the flow of the gradients in our recurrent setting, we tried to add a `stop_gradient` operator to the generated weights that we pass from task to task. Essentially, this means that the HT cannot affect the input weights directly. Curiously, making this modification sometimes improves the results, but we couldn’t find a scenario where this improvement would be consistent enough to be worth reporting in the paper.

---

### Decision · Program_Chairs · 2023-01-20

**Decision:**

Reject

**Justification For Why Not Higher Score:**

The paper is not very novel - its just an adaptation of HT using prototypical loss for few shot incremental learning.
Evaluations do not include comparison with baselines.


**Justification For Why Not Lower Score:**

N/A

**Metareview: Summary, Strengths And Weaknesses:**

The paper proposes an incremental hypertransformer (IHT) to generate weights for incremental task scenario in continual learning. The method is pretty straightforward extension of the HT for continual learning, using a prototypical loss (instead of the cross entropy loss). The method has been evaluated on  1-shot, 20-way OMNIGLOT and 5-shot, 5-way TIEREDIMAGENET.

The IHT is not very novel - it is a direct application of the HT with prototypical loss, enabling IHT to generate weights suited for all tasks (1,..t) using weights for (t-1) known tasks and the data for the t-th task.

The model performance has not been compared with any baselines - while the authors argue that the methods referred by Reviewer UZsQ are not comparable, there are several works on few shot class incremental learning (Specifically) and few shot incremental learning (generally). If not any of these, the performance of IHT could be compared against other CIL/continual learning algorithms.